

# Lorentz symmetry fractionalization and dualities in (2+1)d

**Po-Shen Hsin[1†] and Shu-Heng Shao[2⋆]**

**1** Walter Burke Institute for Theoretical Physics,
California Institute of Technology, Pasadena, CA 91125, USA
**2** School of Natural Sciences, Institute for Advanced Study, Princeton, NJ 08540, USA

† phsin@caltech.edu, ⋆ shao@ias.edu

## Abstract

We discuss symmetry fractionalization of the Lorentz group in (2+1)$d$ non-spin quantum field theory (QFT), and its implications for dualities. We prove that two inequivalent non-spin QFTs are dual as spin QFTs if and only if they are related by a Lorentz symmetry fractionalization with respect to an anomalous $\mathbb{Z}_2$ one-form symmetry. Moreover, if the framing anomalies of two non-spin QFTs differ by a multiple of 8, then they are dual as spin QFTs if and only if they are also dual as non-spin QFTs. Applications to summing over the spin structures, time-reversal symmetry, and level/rank dualities are explored. The Lorentz symmetry fractionalization naturally arises in Chern-Simons matter dualities that obey certain spin/charge relations, and is instrumental for the dualities to hold when viewed as non-spin theories.



# 1   Introduction

Symmetry fractionalization in quantum field theory (QFT) is a general phenomenon where some massive particles (anyons) transform in projective representations of certain zero-form global symmetry $G$, while local operators are in linear representations of $G$. In other words, the massive particles, or more precisely the line operators, carry fractional symmetry charges. The fractional quantum Hall effect is a classic example where the anyons carry fractional $U(1)$ charges (see *e.g.* [1]).

The symmetry fractionalization is particularly interesting when the symmetry group is taken to be the Lorentz group $SO(D)_{\text{Lorentz}}$ in $D$ dimensional Euclidean spacetime. The projective representations of the Lorentz group are classified by $H^2(SO(D)_{\text{Lorentz}}, U(1)) = \mathbb{Z}_2$, where the nontrivial projective representation corresponds to the fermion. An example of Lorentz symmetry fractionalization in (3+1)$d$ is discussed in [2–5] for the pure $U(1)$ gauge theory. The theory has line operators given by combinations of the Wilson lines (electric particle) and the 't Hooft lines (magnetic monopole), and they transform under the $U(1)$ electric and the $U(1)$ magnetic one-form global symmetries [6]. The different fractionalizations of Lorentz symmetry (without time-reversal) correspond to different ways of changing the spin of the particles by $\frac{1}{2}$ [2, 3]. For instance, the following transformation relates different symmetry fractionalizations:

Wilson lines are bosons → Wilson lines with even/odd charges are bosons/fermions .

These different choices correspond to activating different backgrounds of the $U(1) \times U(1)$ one-form symmetry expressed in terms of certain discrete gravitational background fields [4]. Different symmetry fractionalizations can have different 't Hooft anomalies. For instance, in the Maxwell theory with vanishing $\theta$ angle, the theory is known to have a gravitational anomaly if both the basic electric and magnetic particles are fermions (as opposed to bosons) [4,7–10], which originates from the anomaly of the one-form symmetry [6].[1] As we will see later, all of the above features have counterparts in (2+1)$d$.

In this paper, we discuss symmetry fractionalization for the Lorentz group of bosonic/non-spin (2+1)$d$ QFT with a $\mathbb{Z}_2$ one-form symmetry. We will focus on time-preserving Lorentz symmetry, so the theory does not need to be time-reversal invariant. More specifically, the Lorentz symmetry fractionalization is realized by activating a nontrivial $\mathbb{Z}_2$ one-form symmetry background using the Lorentz group background fields. The Lorentz symmetry fractionalization modifies the spins and statistics of the anyons (while leaving the local operator data

---

[1]See [9, 11–13] for related works in (3+1)$d$.

invariant), and defines a nontrivial map **F** from a non-spin QFT to another:

$$\mathbf{F}: \quad \text{non-spin QFT} \to \text{non-spin QFT}. \tag{1.1}$$

We will call **F** the *fractionalization map*. In some special cases, **F** maps the non-spin QFT back to itself and can become a zero-form global symmetry of the theory (see Section 2.2 for the example of the twisted $\mathbb{Z}_2$ gauge theory). If the theory does not have a $\mathbb{Z}_2$ one-form symmetry, then the fractionalization of Lorentz symmetry is unique and there is no non-trivial map **F**.

In $(2+1)d$, non-spin topological quantum field theories (TQFT) are described by modular tensor category [14–17].[2] The data of modular tensor category are characterized by fusions and braidings of the anyons, which obey stringent constraints such as the pentagon and the hexagon identities. The symmetry fractionalization in $(2+1)d$ TQFT has been systematically studied in [18–23]. Applying the Lorentz symmetry fractionalization to a non-spin TQFT, the fractionalization map **F** produces another non-spin TQFT where the spins of some anyons are shifted by $\frac{1}{2}$, while the other TQFT data (such as the fusion algebra and the Hopf braiding of anyons) remain invariant. We will discuss various examples of non-spin TQFTs related by Lorentz symmetry fractionalizations.

The fractionalization map has an interesting connection to dualities between spin and non-spin QFTs. There are examples of QFTs that are dual as spin theories, but inequivalent as non-spin theories. For example, the $\mathbb{Z}_2$ gauge theory $(\mathcal{Z}_2)_0$ and the $Spin(8)_1$ Chern-Simons theory are two such non-spin QFTs. What is the relation between two such non-spin QFTs? In Section 4.1, we prove our main theorem: *two inequivalent non-spin QFTs are dual as spin QFTs if and only if they have a $\mathbb{Z}_2$ one-form symmetry and are related by the corresponding fractionalization map. Moreover, if the framing anomalies of two non-spin QFTs differ by a multiple of 8, then they are dual as spin QFTs if and only if they are also dual as non-spin QFTs.*

We further discuss uplifts of a spin TQFT to non-spin TQFTs in $(2+1)d$. Starting from a spin TQFT, one obtains 16 distinct non-spin TQFTs by summing over the spin structures weighting with different invertible spin TQFTs. We show that the 16 non-spin TQFTs are pairwise related by a fractionalization map, therefore there are only 8 distinct TQFTs when viewed as spin theories. We further explore the implications of our theorem for time-reversal symmetry and level/rank dualities of non-spin TQFTs.

Rather than being a mathematical artifact, the Lorentz symmetry fractionalization arises naturally in Chern-Simons matter dualities in $(2+1)d$. In many infrared dualities, the boson and fermion fields obey certain spin/charge relation in the ultraviolet. In these cases, the dualities can be formulated without choosing a spin structure, despite the appearance of fermion fields in the Lagrangian. The spin/charge relation implies that the gauge bundle in the ultraviolet is twisted by the Lorentz group. In the infrared, this results in a fractionalization map for the TQFTs when viewed as non-spin theories. The presence of the fractionalization map resolves some seeming mismatches of the dualities when viewed as non-spin theories.

The rest of the paper is organized as follows. In Section 2, we define and explore various basic properties of the fractionalization map. Various examples of fractionalization maps on TQFTs are presented in Section 2 and 3. In Section 4, we prove our main theorem on the relation between spin dualities and the fractionalization map. We apply our theorem to time-reversal symmetry and level/rank dualities for non-spin TQFTs. In Section 5, we discuss the implication of the fractionalization map for non-spin Chern-Simons matter dualities. Appendix A reviews the $\mathbb{Z}_N$ gauge theories in $(2+1)d$. In Appendix B we discuss the relation between the spin duality map and the $\mathbb{Z}_2$ one-form symmetry. In Appendix C we discuss another map between non-spin QFTs related to different ways of summing over the spin structures in $(1+1)d$ and $(2+1)d$.

---

[2]We will ignore trivial non-spin TQFTs, whose framing anomalies are multiples of 8. They correspond to bosonic gravitational Chern-Simons terms and thus do not contribute to the dynamics in $3d$.

## 2 Symmetry fractionalization map

We start with a brief review of symmetry fractionalization in $(2+1)d$. Consider a $(2+1)d$ QFT with a zero-form global symmetry $G$ and a one-form global symmetry $\mathcal{A}$. Symmetry fractionalization in $(2+1)d$ is a phenomenon where the line operators can be in projective representations of $G$, while local operators are in linear representations of $G$. More specifically, the symmetry fractionalization can be realized by activating the $\mathcal{A}$ background field $B$ using the pullback of an element in $H^2(G, \mathcal{A})$ by the $G$ background field [11]. This nontrivial background for the one-form symmetry inserts the symmetry generator – Abelian anyon– at the junction of three $G$ defects as specified by the chosen element in $H^2(G, \mathcal{A})$ [18, 20–22].

The two-form background $B$ attaches those lines that transform under the one-form symmetry with the "Wilson surface" $\oint B$. For a line with one-form symmetry charge $q \in \widehat{\mathcal{A}} = \text{Hom}(\mathcal{A}, U(1))$, the symmetry fractionalization specified by $\eta \in H^2(G, \mathcal{A})$ attaches the line with an additional surface that lives the $(1+1)d$ symmetry-protected topological (SPT) phase $q(\eta) \in H^2(G, U(1))$. From the anomaly inflow mechanism, the SPT phase implies that the anyon on the line operator acquires an additional projective representation of the $G$ symmetry as specified by $q(\eta)$.

### 2.1 Lorentz symmetry fractionalization

Consider a non-spin QFT $\mathcal{T}$ with an one-form symmetry $\mathcal{A} = \mathbb{Z}_2$, generated by the anyon $a$. Instead of taking the zero-form symmetry $G$ to be an internal symmetry, we will consider the case when $G$ is the bosonic Lorentz group $SO(3)_{\text{Lorentz}}$. It follows that different symmetry fractionalizations are classified by $H^2(G, \mathcal{A}) = H^2(SO(3)_{\text{Lorentz}}, \mathbb{Z}_2) = \mathbb{Z}_2[w_2]$ where $w_2$ is the second Stiefel-Whitney class of the Lorentz bundle.

To change the symmetry fractionalization, we activate the two-form background field $B$ for the $\mathbb{Z}_2$ one-form symmetry using the background field of the bosonic Lorentz group $SO(3)_{\text{Lorentz}}$:

$$B = w_2 \,. \tag{2.1}$$

While the nontrivial background (2.1) does not change the spectrum and correlation functions of local operators, it does modify the quantum numbers of the line defects, or the anyons. The line operators carrying the $\mathbb{Z}_2$ charges now acquire additional projective representations of the bosonic Lorentz group $SO(3)_{\text{Lorentz}}$, i.e. the spins of the particles are shifted by $\frac{1}{2}$.

Explicitly, the spin $h$ of an anyon $b$ is shifted by

$$h[b] \to h[b] + \frac{q_a[b]}{2} \mod 1$$
$$= \begin{cases} \text{invariant,} & \text{if } b \text{ is } \mathbb{Z}_2 \text{ even} \\ \text{changed by } \frac{1}{2}, & \text{if } b \text{ is } \mathbb{Z}_2 \text{ odd} \end{cases} \,, \tag{2.2}$$

where $q_a[b] = 0, 1 \mod 2$ is the charge of the anyon $b$ under the $\mathbb{Z}_2$ one-form symmetry generated by $a$.[3] The fusion rules, $F$-symbols, Hopf braiding and other correlation functions (except those that detect the spin of particles) are the same for different Lorentz symmetry fractionalizations. Hence, the Lorentz symmetry fractionalization defines a map from one non-spin QFT $\mathcal{T}$ to another non-spin QFT, which will be denoted as $\mathbf{F}_a[\mathcal{T}]$. We will call this

---

[3]Our convention for the charge is that if an anyon $b$ has $\mathbb{Z}_2$ one-form charge $q$, then it transforms under the non-trivial element of the $\mathbb{Z}_2$ one-form symmetry by a phase $(-1)^q$. The $\mathbb{Z}_2$ charge is fixed by the spins as

$$q_a[b] = 2(h[b] + h[a] - h[ab]) \mod 2 \,, \tag{2.3}$$

where $ab$ is the fusion of $a$ and $b$.

map the *fractionalization map* with respect to a $\mathbb{Z}_2$ one-form symmetry $\mathcal{A}$ generated by $a$. The operational definition of the fractionalization map is given in (2.2).

In theory with general one-form symmetry $\mathcal{A}$, the classification of Lorentz symmetry fractionalizations is $H^2(SO(3)_{\text{Lorentz}}, \mathcal{A}) = \prod_i \mathbb{Z}_2^{(i)}[w_2]$ with $i$ labelling the independent $\mathbb{Z}_2$ generators in $\mathcal{A}$. The classification corresponds to turning on backgrounds $B^{(i)} = w_2$ for different $\mathbb{Z}_2$ subgroups of the one-form symmetry $\mathcal{A}$. From the definition (2.1), it is clear that the fractionalization map is a homomorphism with respect to the one-form symmetry. Explicitly, let $a_1$ and $a_2$ be two $\mathbb{Z}_2$ one-form symmetries of $\mathcal{T}$, then

$$\mathbf{F}_{a_2} \circ \mathbf{F}_{a_1} = \mathbf{F}_{a_2 a_1}. \tag{2.4}$$

The $a_2 \in \mathbf{F}[\mathcal{T}]$ anyon on the left hand side is the image of $a_2 \in \mathcal{T}$ under the fractionalization map.

Throughout this paper, the dualities between non-spin theories hold up to some invertible non-spin TQFTs (such as $(E_8)_1$) whose framing anomaly is a multiple of 8. Such invertible non-spin TQFTs are equivalent to bosonic gravitational Chern-Simons terms $16n\text{CS}_{\text{grav}}$ for some integer $n$, and thus do not affect the $3d$ dynamics. For this reason, we will only consider the framing anomaly $c$ mod 8.

## 2.2  Examples: $\mathbb{Z}_2$ gauge theories

Before we embark on a general discussion of the fractionalization map, we start with a couple of simple examples of fractionalization maps of non-spin TQFTs.

**Untwisted $\mathbb{Z}_2$ gauge theory $(\mathcal{Z}_2)_0 = Spin(16)_1$**  Consider the untwisted $\mathbb{Z}_2$ gauge theory $(\mathcal{Z}_2)_0$, viewed as a non-spin TQFT. Note that the untwisted $\mathbb{Z}_2$ gauge theory can also be realized as the $Spin(16)_1$ Chern-Simons theory. There are four anyons: $1, f, e, m$ with fusion rules:

$$f \times f = e \times e = m \times m = 1, \quad e \times m = f, \quad m \times f = e, \quad f \times e = m. \tag{2.5}$$

(In the convention of Appendix A, $e = W_{1,0}$, $m = W_{0,1}$, and $f = W_{1,1}$.) There are three one-form symmetries $\mathbb{Z}_2^{(f)}, \mathbb{Z}_2^{(e)}, \mathbb{Z}_2^{(m)}$ generated by $f, e, m$, respectively. The spins and the $\mathbb{Z}_2$ charges are listed below

$$(\mathcal{Z}_2)_0 = Spin(16)_1:$$

|  | 1 | $f$ | $e$ | $m$ |
|---|---|---|---|---|
| $h$ | 0 | $\frac{1}{2}$ | 0 | 0 |
| $q_f$ | 0 | 0 | 1 | 1 |
| $q_e$ | 0 | 1 | 0 | 1 |
| $q_m$ | 0 | 1 | 1 | 0 |

$(c = 0 \bmod 8). \tag{2.6}$

Here $q_e = 0, 1 \bmod 2$ is the charge with respect to $\mathbb{Z}_2^{(e)}$, and so on.

Now we can apply the fractionalization map (2.2) with respect to each of the three $\mathbb{Z}_2$ one-form symmetries. The fractionalization map with respect to $\mathbb{Z}_2^{(e)}$ maps $(\mathcal{Z}_2)_0$ back to itself, but exchanging the anyons $f$ and $m$. Similarly, the fractionalization map with respect to $\mathbb{Z}_2^{(m)}$ maps $(\mathcal{Z}_2)_0$ to itself and exchanges the anyons $f$ and $e$. The more interesting map is the one with respect to $\mathbb{Z}_2^{(f)}$: it maps $(\mathcal{Z}_2)_0$ to $Spin(8)_1$. The anyons and their spins of the $Spin(8)_1$ Chern-Simons theory are

$$Spin(8)_1:$$

|  | 1 | $f$ | $e$ | $m$ |
|---|---|---|---|---|
| $h$ | 0 | $\frac{1}{2}$ | $\frac{1}{2}$ | $\frac{1}{2}$ |
| $q_f$ | 0 | 0 | 1 | 1 |
| $q_e$ | 0 | 1 | 0 | 1 |
| $q_m$ | 0 | 1 | 1 | 0 |

$(c = 4 \bmod 8). \tag{2.7}$

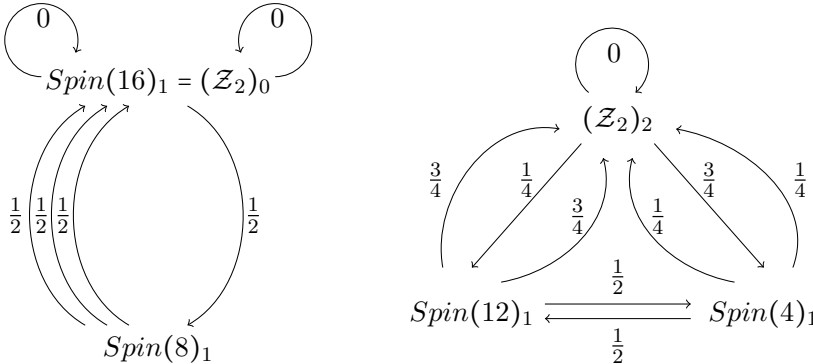

Figure 1: The fractionalization maps of $\mathbb{Z}_2$ gauge theories. The number next to the arrow labels the spin of the $\mathbb{Z}_2$ line used for the fractionalization map.

The fusion rules of $Spin(8)_1$ are again given by (2.5).

To summarize:

$$
\begin{aligned}
\mathbf{F}_f[(\mathcal{Z}_2)_0] &\longleftrightarrow Spin(8)_1\,, \\
\mathbf{F}_{e,m}[(\mathcal{Z}_2)_0] &\longleftrightarrow (\mathcal{Z}_2)_0\,.
\end{aligned}
\tag{2.8}
$$

Conversely, we can start with $Spin(8)_1$, and apply the fractionalization map with respect to its one-form symmetries $\mathbb{Z}_2^{(f)}, \mathbb{Z}_2^{(e)}, \mathbb{Z}_2^{(m)}$. The $Spin(8)_1$ theory has an $\mathbb{S}_3$ zero-form symmetry permuting the three anyons $f, e, m$. Therefore, the fractionalization maps with respect to the three one-form symmetries are identical, which take $Spin(8)_1$ back to $(\mathcal{Z}_2)_0$:

$$
\mathbf{F}_{f,e,m}[Spin(8)_1] \longleftrightarrow (\mathcal{Z}_2)_0\,.
\tag{2.9}
$$

See the left figure of Figure 1 for the actions of the fractionalization maps.

**Twisted $\mathbb{Z}_2$ gauge theory** $(\mathcal{Z}_2)_2 = U(1)_2 \times U(1)_{-2}$   Consider the non-spin $\mathbb{Z}_2$ gauge theory $(\mathcal{Z}_2)_2$ with a Dijkgraaf-Witten twist [24]. It is equivalent to the semion-antisemion theory $U(1)_2 \times U(1)_{-2}$. We will label the lines $W_{n_e, n_m}$ by their electric and magnetic charges $n_e$ and $n_m$ (see Appendix A for our conventions). There are four anyons, the trivial line $1 = W_{0,0}$, the electric line $W_{1,0}$, the magnetic line $W_{0,1}$, and the dyonic line $W_{1,1}$. The fusion rules are $W_{n_e, n_m} W_{n'_e, n'_m} = W_{n_e + n'_e, n_m + n'_m}$, which is the same as (2.5). The spins and the $\mathbb{Z}_2$ charges are

$(\mathcal{Z}_2)_2$ :

| $(n_e, n_m)$ | $(0,0)$ | $(1,1)$ | $(1,0)$ | $(0,1)$ |
|---|---|---|---|---|
| $h$ | 0 | $\frac{1}{4}$ | 0 | $\frac{3}{4}$ |
| $q_{(1,1)}$ | 0 | 1 | 1 | 0 |
| $q_{(1,0)}$ | 0 | 1 | 0 | 1 |
| $q_{(0,1)}$ | 0 | 0 | 1 | 1 |

$(c = 0 \bmod 8)$.   (2.10)

The fractionalization maps with respect to the three one-form symmetries $\mathbb{Z}_2^{(1,1)}, \mathbb{Z}_2^{(1,0)}, \mathbb{Z}_2^{(0,1)}$ are

$$
\begin{aligned}
\mathbf{F}_{(1,1)}[(\mathcal{Z}_2)_2] &\longleftrightarrow Spin(12)_1\,, \\
\mathbf{F}_{(1,0)}[(\mathcal{Z}_2)_2] &\longleftrightarrow (\mathcal{Z}_2)_2\,, \\
\mathbf{F}_{(0,1)}[(\mathcal{Z}_2)_2] &\longleftrightarrow Spin(4)_1\,.
\end{aligned}
\tag{2.11}
$$

The $Spin(N)_1$ TQFT with $N = 0$ mod 4 has four anyons $1, f, e, m$ with fusion rules (2.5). The spins and their charges with respect to the one-form symmetries $\mathbb{Z}_2^{(f)}, \mathbb{Z}_2^{(e)}, \mathbb{Z}_2^{(m)}$ are

$$Spin(N)_1 \ (N = 0 \text{ mod } 4): \quad
\begin{array}{|c|c|c|c|c|}
\hline
 & 1 & f & e & m \\
\hline
h & 0 & \frac{1}{2} & \frac{N}{16} & \frac{N}{16} \\
\hline
q_f & 0 & 0 & 1 & 1 \\
\hline
q_e & 0 & 1 & \frac{N}{4} & \frac{N}{4} - 1 \\
\hline
q_m & 0 & 1 & \frac{N}{4} - 1 & \frac{N}{4} \\
\hline
\end{array}
\quad (c = \frac{N}{2} \text{ mod } 8). \quad (2.12)$$

Note that the fractionalization map with respect to $m$ exchanges the spin $\frac{1}{4}$ line with the spin $-\frac{1}{4}$ line, which is the time-reversal symmetry of $(\mathcal{Z}_2)_2$.

Conversely, the fractionalization maps of $Spin(4)_1$ and $Spin(12)_1$ are

$$\begin{aligned}
\mathbf{F}_f[Spin(4)_1] &\longleftrightarrow Spin(12)_1, & \mathbf{F}_f[Spin(12)_1] &\longleftrightarrow Spin(4)_1, \\
\mathbf{F}_{e,m}[Spin(4)_1] &\longleftrightarrow \mathbf{F}_{e,m}[Spin(12)_1] &\longleftrightarrow (\mathcal{Z}_2)_2.
\end{aligned} \quad (2.13)$$

See the right figure of Figure 1 for the actions of the fractionalization maps.

## 2.3 Framing anomaly

Since the symmetry fractionalization activates the one-form symmetry background $\mathcal{A}$, the anomaly of the one-form symmetry gives rise to an anomaly of the zero-form symmetry $G$ through symmetry fractionalizations [11]. In the case of the Lorentz symmetry fractionalization (2.1), the one-form symmetry anomaly gives rise to the framing anomaly. Here we discuss the change of the framing anomaly under the fractionalization map.

Suppose theory $\mathcal{T}$ has a $\mathbb{Z}_2$ one-form symmetry, then the symmetry is generated by a spin $\frac{p}{4}$ line for some integer $p$, and the 't Hooft anomaly of the one-form symmetry is captured by the (3+1)$d$ SPT term: [25]

$$2\pi \frac{p}{4} \int_{M_4} \mathcal{P}(B), \quad (2.14)$$

where $B$ is the two-form background $\mathbb{Z}_2$ gauge field and $\mathcal{P}$ is the Pontryagin square operation [26] (see, for example, [11, 27, 28] for physics reviews). If we set $B = w_2(SO(3)_{\text{Lorentz}})$, the (3+1)$d$ SPT becomes $2\pi \frac{p}{4} \int_{M_4} \mathcal{P}(w_2)$. The latter is related to the first Pontryagin class $p_1$ by [29]

$$p_1 = \mathcal{P}(w_2) + 2w_4 \text{ mod } 4, \quad (2.15)$$

where $2w_4$ is regarded as a mod 4 class via the inclusion map $\mathbb{Z}_2 \hookrightarrow \mathbb{Z}_4$. It is known that $w_1^4 + w_2^2 + w_4 = 0$ on any closed four-manifold [30]. On an oriented four-manifold, we then have

$$\mathcal{P}(w_2) = -p_1 \text{ mod } 4. \quad (2.16)$$

Hence under the fractionalization map, $\mathbf{F}[\mathcal{T}]$ gains the following (3+1)$d$ SPT for the framing anomaly:

$$-2\pi \frac{p}{4} \int_{M_4} p_1 = -\frac{p}{48\pi} \int_{M_4} \text{tr} R \wedge R. \quad (2.17)$$

In other words, the framing anomaly is changed by

$$\Delta c \equiv c(\mathbf{F}[\mathcal{T}]) - c(\mathcal{T}) = -2p \text{ mod } 8. \quad (2.18)$$

## 2.4 General case

Consider a $3d$ non-spin theory $\mathcal{T}$ with a $\mathbb{Z}_2$ one-form symmetry generated by a symmetry line operator of spin $\frac{p}{4}$ mod 1 for some integer $p$ [25]. The $\mathbb{Z}_2$ gauge theory $(\mathcal{Z}_2)_{-2p}$ has a $\mathbb{Z}_2$ one-form symmetry line whose spin is $-\frac{p}{4}$ mod 1. The latter acts nontrivially on lines carrying electric charges. We can therefore gauge the diagonal $\mathbb{Z}_2$ one-form symmetry of $\mathcal{T} \times (\mathcal{Z}_2)_{-2p}$. The gauged theory is dual to $\mathcal{T}$ [25, 31]

$$\mathcal{T} \quad \longleftrightarrow \quad \frac{\mathcal{T} \times (\mathcal{Z}_2)_{-2p}}{\mathbb{Z}_2} . \tag{2.19}$$

The $\mathbb{Z}_2$ one-form symmetry on the left hand side is described on the right hand side by the $\mathbb{Z}_2$ one-form symmetry generated by the magnetic line in the $\mathbb{Z}_2$ gauge theory $(\mathcal{Z}_2)_{-2p}$, which also has spin $\frac{p}{4}$ mod 1 (since it has zero electric charge, this line survives the $\mathbb{Z}_2$ quotient on the right hand side).

If we apply the fractionalization map on (2.19), it acts on the right hand side using the spin $\frac{p}{4}$ line of $(\mathcal{Z}_2)_{-2p}$, which gives a closed form expression for $\mathbf{F}[\mathcal{T}]$:

$$\mathbf{F}[\mathcal{T}] \quad \longleftrightarrow \quad \frac{\mathcal{T} \times Spin(-4p)_1}{\mathbb{Z}_2^{(\mathbf{s})}} , \tag{2.20}$$

where we have used $\mathbf{F}[(\mathcal{Z}_2)_{-2p}] \longleftrightarrow Spin(-4p)_1$ from Section 2.2. The gauged one-form symmetry $\mathbb{Z}_2^{(\mathbf{s})}$ is generated by the tensor product of the spin $\frac{p}{4}$ line of $\mathcal{T}$ and the spin $-\frac{p}{4}$ line in the spinor representation of $Spin(-4p)_1$. The superscript $(\mathbf{s})$ is to remind the reader that the $\mathbb{Z}_2$ symmetry involves the line in the spinor representation of $Spin(-4p)_1$. Note that $Spin(-4p)_1 \longleftrightarrow Spin(4p)_{-1}$ (up to trivial TQFTs of $c \in 8\mathbb{Z}$ such as $(E_8)_1$). Indeed, the chiral central charge of the right hand side is $c(\mathcal{T}) - 2p$, consistent with the general rule (2.18).

Let us comment on some properties of the fractionalization map using the expression (2.20):

- The fractionalization map $\mathbf{F}$ leaves the theory invariant if and only if the $\mathbb{Z}_2$ one-form symmetry is generated by a line of integer spin (*i.e.* if the one-form symmetry is non-anomalous [25]):

$$\mathbf{F}[\mathcal{T}] \quad \longleftrightarrow \quad \mathcal{T}, \quad (p = 0). \tag{2.21}$$

This can be seen from (2.20) using $\mathcal{T} = \frac{\mathcal{T} \times (\mathcal{Z}_2)_0}{\mathbb{Z}_2}$.

- When the theory has multiple $\mathbb{Z}_2$ one-form symmetries, the fractionalization maps with respect to $\mathbb{Z}_2$ lines of different spins produce different theories. Since the spin can take at most four distinct values, the fractionalization map can at most connect 4 non-spin theories. In Figure 1 we have shown examples of fractionalization maps that connect 2 and 3 different non-spin theories.

- The $\mathbb{Z}_2^{(\mathbf{s})}$ quotient on the right hand side of (2.20) gauges the diagonal one-form symmetry that acts non-trivially on the fermion line $f$ of $Spin(-4p)_1$ in the vector representation. Thus $\mathbf{F}[\mathcal{T}]$ is related to $\mathcal{T}$ by attaching the $\mathbb{Z}_2^{(\mathbf{s})}$ odd lines in $\mathcal{T}$ with the fermion line $f$ of $Spin(-4p)_1$ in the vector representation. The correlation functions of the fermion lines $f$ are trivial (except for the dependence on the framing of the lines [32]). It follows that the correlation functions of $\mathbf{F}[\mathcal{T}]$ can only differ from $\mathcal{T}$ by the statistics of the lines.

### 2.5 Map on the 2d chiral algebras

$3d$ TQFTs are associated with $2d$ chiral algebras. It is then natural to ask what is the operation on the chiral algebra that corresponds to the fractionalization map. The $\mathbb{Z}_2$ one-form symmetry in $3d$ corresponds to a $\mathbb{Z}_2$ simple current in the chiral algebra, *i.e.* primary operators with Abelian fusion algebra [33–35]. Gauging a one-form symmetry in $3d$ corresponds to an extension of the chiral algebra by the $\mathbb{Z}_2$ simple current [36, 37]. Thus, using (2.20), the fractionalization map induces the following operation on the chiral algebra: first we tensor it with the chiral algebra of $Spin(-4p)_1$ ($4p$ right-moving $2d$ Majorana fermions), and then take the $\mathbb{Z}_2$ extension of the tensor product chiral algebra. Note that there can be multiple $2d$ chiral algebras that correspond to the same $3d$ TQFT. Here we only describe a map that corresponds to the $3d$ fractionalization map.

## 3 More examples

### 3.1 $U(1)_{\pm 2}$ Chern-Simons theory

Consider the $U(1)_2$ Chern-Simons theory, viewed as a non-spin TQFT. There are only two anyons $1, s$, with $s \times s = 1$ generating a $\mathbb{Z}_2$ one-form symmetry. Their spins and $\mathbb{Z}_2$ charges $q_s$ are

$$
U(1)_2 : \quad
\begin{array}{|c|c|c|}
\hline
 & 1 & s \\
\hline
h & 0 & \frac{1}{4} \\
\hline
q_s & 0 & 1 \\
\hline
\end{array}
\quad (c = 1 \bmod 8) .
\tag{3.1}
$$

The fractionalization map modifies the spin of $s$ from $\frac{1}{4}$ to $\frac{1}{4} + \frac{1}{2} = \frac{3}{4}$. We therefore end up with the $U(1)_{-2}$ Chern-Simons theory:

$$
U(1)_{-2} : \quad
\begin{array}{|c|c|c|}
\hline
 & 1 & \bar{s} \\
\hline
h & 0 & \frac{3}{4} \\
\hline
q_{\bar{s}} & 0 & 1 \\
\hline
\end{array}
\quad (c = -1 \bmod 8) .
\tag{3.2}
$$

Conversely, the fractionalization map of $U(1)_{-2}$ is $U(1)_2$. In summary,

$$
\mathbf{F}_s[U(1)_2] \quad \longleftrightarrow \quad U(1)_{-2} , \qquad \mathbf{F}_{\bar{s}}[U(1)_{-2}] \quad \longleftrightarrow \quad U(1)_2 .
\tag{3.3}
$$

### 3.2 $Spin(N)_1$ Chern-Simons theory

Let us first review the anyons in the (non-spin) $Spin(N)_1$ Chern-Simons theory. See, for example, Appendix C of [38] or Table 1-3 of [15] for reviews. The TQFT is described by $N$ chiral Majorana edge fermions in $2d$.

- If $N$ is odd, there are three anyons $1, f, \sigma$ with fusion rules:

$$
f \times f = 1 , \quad f \times \sigma = \sigma \times f = \sigma , \quad \sigma \times \sigma = 1 + f .
\tag{3.4}
$$

  There is a $\mathbb{Z}_2$ one-form symmetry generated by the spin $\frac{1}{2}$ line $f$. The spins and the $\mathbb{Z}_2$ charges of the anyons are

$$
Spin(N)_1 \quad (N : \text{odd}) \quad
\begin{array}{|c|c|c|c|}
\hline
 & 1 & f & \sigma \\
\hline
h & 0 & \frac{1}{2} & \frac{N}{16} \\
\hline
q_f & 0 & 0 & 1 \\
\hline
\end{array}
\tag{3.5}
$$

  For example, $Spin(1)_1$ is the non-spin Ising TQFT, and $Spin(3)_1 = SU(2)_2$.

- If $N = 0$ mod 4, there are four anyons $1, f, e, m$. The one-form symmetry is $\mathbb{Z}_2 \times \mathbb{Z}_2$. The spins and the $\mathbb{Z}_2$ charges are given in (2.12). The fusion rules are given in (2.5). For example, $Spin(16)_1 = (\mathcal{Z}_2)_0$.

- If $N = 2$ mod 4, there are four anyons $1, f, a, \bar{a}$, obeying the fusion rules

$$f \times f = a \times \bar{a} = 1, \quad a \times f = \bar{a}, \quad \bar{a} \times f = a, \quad a \times a = \bar{a} \times \bar{a} = f. \tag{3.6}$$

The one-form symmetry is $\mathbb{Z}_4$. The spins and the charges of the $\mathbb{Z}_2$ one-form symmetry subgroup (generated by $f$) are

$$Spin(N)_1 \quad (N = 2 \text{ mod } 4): \quad \begin{array}{|c|c|c|c|c|} \hline & 1 & f & a & \bar{a} \\ \hline h & 0 & \frac{1}{2} & \frac{N}{16} & \frac{N}{16} \\ \hline q_f & 0 & 0 & 1 & 1 \\ \hline \end{array}. \tag{3.7}$$

For example, $Spin(2)_1 = U(1)_4$.

**Fractionalization map** For any $N$, the fractionalization map with respect to the spin $\frac{1}{2}$ line $f$ is

$$\mathbf{F}_f[Spin(N)_1] \quad \longleftrightarrow \quad Spin(N+8)_1. \tag{3.8}$$

The chiral central charge of $Spin(N)_1$ is $c = \frac{N}{2}$ mod 8, which is shifted as (2.18) under the fractionalization map. If $N = 0$ mod 4, the one-form symmetry is $\mathbb{Z}_2 \times \mathbb{Z}_2$, so there are other $\mathbb{Z}_2$ symmetries that we can perform the fractionalization map. This was discussed in (2.13).

# 4 Application I: TQFT duality

In this section we apply the fractionalization map to dualities between spin and non-spin TQFTs. We also discuss implications of the fractionalization maps for time-reversal symmetry and level/rank dualities.

Since the Lorentz fractionalization map only changes the line operators data but not those of the local operators, its action is most drastic in TQFTs. For this reason we will focus on TQFTs in this section. However we emphasize that the discussions and conclusions of this section can be generalized straightforwardly to the case of general bosonic QFTs in $3d$, and the dualities can be infrared dualities instead of exact dualities. In the generalization to bosonic quantum field theory, the framing anomaly in the following discussions can be defined by the coefficient of the parity-odd contact term in the stress tensor two-point function [39, 40]. This coefficient can only be changed by a multiple of 8 by adding the bosonic gravitational Chern-Simons term $16n\text{CS}_{\text{grav}}$ with integer $n$.

## 4.1 Lifting spin dualities with the fractionalization map

A non-spin QFT gives rise to a spin QFT by tensoring with the invertible spin TQFT $\{1, f\}$ where $f$ is the transparent fermion line of spin $\frac{1}{2}$. The TQFT $\{1, f\}$ can be described by the spin Chern-Simons theory $SO(L)_1$, whose chiral central charge $c = L/2$ depends on $L$ but not the line operators. It can also be expressed as the gravitational Chern-Simons term $SO(L)_1 \longleftrightarrow -L\text{CS}_{\text{grav}}$ and the transparent fermion line $f$ (in the vector representation of $SO(L)$) is identified with a gravitational line. In particular, $SO(L)_1 \times SO(L')_1 \longleftrightarrow SO(L+L')_1 \longleftrightarrow -(L+L')\text{CS}_{\text{grav}}$.

**Lemma 1**  Suppose two non-spin QFTs $\mathcal{T}_1, \mathcal{T}_2$ are dual as spin QFTs, *i.e.*

$$\mathcal{T}_1 \times SO(0)_1 \quad \longleftrightarrow \quad \mathcal{T}_2 \times SO(L)_1 \, , \tag{4.1}$$

where $L = 2\Delta c = 2(c(\mathcal{T}_1) - c(\mathcal{T}_2))$ to balance the difference in chiral central charge of $\mathcal{T}_1, \mathcal{T}_2$. We will show the following:

- $\Delta c \in 2\mathbb{Z}$.

- If $\Delta c \notin 8\mathbb{Z}$, then $\mathcal{T}_1, \mathcal{T}_2$ must have a $\mathbb{Z}_2$ one-form global symmetry generated by a line of spin $\pm \Delta c/8$, respectively.[4]

With Lemma 1, we prove our main theorem:

**Theorem 1**  Let $\mathcal{T}_1, \mathcal{T}_2$ be two non-spin QFTs with $\Delta c = c(\mathcal{T}_1) - c(\mathcal{T}_2)$.

- When $\Delta c \notin 8\mathbb{Z}$, the two non-spin QFTs $\mathcal{T}_1, \mathcal{T}_2$ are dual as spin QFTs (4.1) if and only if

$$\mathbf{F}[\mathcal{T}_1] \longleftrightarrow \mathcal{T}_2, \quad \mathcal{T}_1 \longleftrightarrow \mathbf{F}[\mathcal{T}_2] \, , \tag{4.2}$$

  with respect to the $\mathbb{Z}_2$ one-form symmetries in Lemma 1.

- When $\Delta c \in 8\mathbb{Z}$, $\mathcal{T}_1, \mathcal{T}_2$ are dual as spin QFTs (4.1) if and only if they are dual as non-spin QFTs, *i.e.* $\mathcal{T}_1 \longleftrightarrow \mathcal{T}_2$.

The proof relies on summing over the spin structures in a spin QFT (see Section 4.2 for more details). For our application, it is sufficient to know the result for the spin Chern-Simons theory $SO(M)_1$. In $SO(M)_1$, changing the spin structure by a classical $\mathbb{Z}_2$ gauge field can be identified with changing the background for the $\mathbb{Z}_2$ magnetic symmetry generated by $\exp(i\pi \oint w_2^{SO(M)})$ [31, 41]. Thus summing over the spin structures in $SO(M)_1$ produces the non-spin Chern-Simons theory $Spin(M)_1$. In particular, the fermion line of $SO(M)_1$ in the vector representation also belongs to the lines in $Spin(M)_1$, while there are new lines of $Spin(M)_1$ in the spinor representations (for even $M$ there are two such lines related by charge conjugation, and they have spin $\frac{M}{16}$ mod 1).

Summing over the spin structures in the duality (4.1) (without tensoring with extra invertible spin TQFT $SO(r)_1$) produces the following duality for non-spin QFT:

$$\mathcal{T}_1 \times (\mathcal{Z}_2)_0 \quad \longleftrightarrow \quad \mathcal{T}_2 \times Spin(L)_1 \, , \tag{4.3}$$

where we used the property that $\mathcal{T}_1, \mathcal{T}_2$ themselves are independent of the choice of the spin structure.

Let us match the one-form symmetry in the dualities (4.1) and (4.3). Since $SO(M)_1$ has the $\mathbb{Z}_2$ fusion algebra for any $M$, we learn that $\mathcal{T}_1, \mathcal{T}_2$ must have the same one-form symmetry. On the other hand, the one-form symmetry of $Spin(M)_1$ depends on $M$ mod 4.[5] Thus matching the one-form symmetry in the duality (4.3) implies $L = 0$ mod 4. It follows that the difference between framing anomalies of $\mathcal{T}_1, \mathcal{T}_2$ is an even integer:

$$\Delta c = c(\mathcal{T}_1) - c(\mathcal{T}_2) = L/2 \in 2\mathbb{Z} \, . \tag{4.5}$$

---

[4] When $\Delta c \in 8\mathbb{Z}$, there is a $\mathbb{Z}_2$ one-form symmetry generated by an integer spin line if and only if the duality map in (4.1) mixes with the transparent line of $SO(r)_1$. See Appendix B.

[5] The one-form symmetry for $Spin(M)_1$ is given by the center of $Spin(M)$:

$$\mathcal{A}(Spin(M)_1) = \begin{cases} \mathbb{Z}_2 & \text{odd } M \\ \mathbb{Z}_4 & M = 2 \text{ mod } 4 \\ \mathbb{Z}_2 \times \mathbb{Z}_2 & M = 0 \text{ mod } 4 \end{cases} \, . \tag{4.4}$$

The duality (4.3) has additional line operators in comparison with the duality (4.1) from $(\mathcal{Z}_2)_0$ and $Spin(L)_1$. In particular, $(\mathcal{Z}_2)_0$ has an electric line $e$ of integer spin that generates a $\mathbb{Z}_2$ one-form symmetry. On the other hand, $Spin(L)_1$ has four lines, two of them are in $SO(L)_1$ and the other two have spin $\frac{L}{16}$ mod 1. Thus in order to match the generator of the one-form symmetry on both sides, if $L \notin 16\mathbb{Z}$ then theory $\mathcal{T}_2$ must have a $\mathbb{Z}_2$ one-form symmetry generated by a line of spin $-\frac{L}{16}$ mod 1:

$$\Delta c = \frac{L}{2} \notin 8\mathbb{Z} \;\Rightarrow\; \exists \text{ a } \mathbb{Z}_2 \text{ line in } \mathcal{T}_2 \text{ of spin } h = -\frac{\Delta c}{8} \quad \text{mod 1} . \tag{4.6}$$

Denoting the spin by $\frac{p}{4}$, this corresponds to $p = -\Delta c/2 = -L/4$ mod 4. Similarly, if $\Delta c \notin 8\mathbb{Z}$ then $\mathcal{T}_1$ must have a $\mathbb{Z}_2$ one-form symmetry generated by a line of spin $\Delta c/8$ mod 1. This concludes the proof for Lemma 1.

We proceed to prove Theorem 1. We can gauge the $\mathbb{Z}_2$ one-form symmetry generated by the integer spin electric line of $(\mathcal{Z}_2)_0$ in (4.3) and produce a new duality

$$\mathcal{T}_1 \quad \longleftrightarrow \quad \frac{\mathcal{T}_2 \times Spin(L)_1}{\mathbb{Z}_2} , \tag{4.7}$$

where we used the property that gauging the $\mathbb{Z}_2$ Wilson line in $(\mathcal{Z}_2)_0$ makes the theory trivial.

When $\Delta c = L/2 \notin 8\mathbb{Z}$, the quotient on the right hand side gauges a diagonal one-form symmetry, whose generator is the product of the line in $\mathcal{T}_2$ of spin $-\frac{L}{16}$ mod 1 and the line in $Spin(L)_1$ of spin $\frac{L}{16}$ mod 1. Thus from (2.20) and $Spin(L)_1 \longleftrightarrow Spin(16-(-L))_1$, the duality (4.7) is

$$\mathcal{T}_1 \quad \longleftrightarrow \quad \mathbf{F}_a[\mathcal{T}_2] , \tag{4.8}$$

where $a$ is the line in $\mathcal{T}_2$ of spin $-\frac{L}{16}$ that generates $\mathbb{Z}_2$ one-form symmetry.

When $\Delta c = L/2 \in 8\mathbb{Z}$, the theory $\mathcal{T}_2$ may or may not have a $\mathbb{Z}_2$ one-form symmetry generated by a line of integer spin (see Appendix B). Since $Spin(L)_1 \longleftrightarrow (\mathcal{Z}_2)_0$, using the duality (2.19) we find that in both cases

$$\mathcal{T}_1 \quad \longleftrightarrow \quad \mathcal{T}_2 , \quad (\Delta c \in 8\mathbb{Z}) . \tag{4.9}$$

In particular, if $\mathcal{T}_2$ has a line $a$ of integer spin, then (4.8) reduces to (4.9) since $\mathbf{F}_a[\mathcal{T}_2] \longleftrightarrow \mathcal{T}_2$. Therefore, assuming the spin duality (4.1), we find that the non-spin QFTs are themselves dual if and only if their framing anomaly differs by a multiple of 8. It is also necessary since an invertible non-spin TQFT has framing anomaly a multiple of 8.

## 4.2 Summing over the spin structures vs. gauging $(-1)^F$

Above we have discussed how a non-spin TQFT gives rise to a spin TQFT by tensoring with an invertible spin TQFT $SO(r)_1$. Here we discuss the opposite process of uplifting a spin TQFT to non-spin TQFTs (see, for example, Appendix C of [38] for a review). As we will see, the resulting non-spin TQFTs are pairwise related by the fractionalization map $\mathbf{F}$.[6]

Starting with a spin TQFT $\widetilde{\mathcal{T}}$, there are 16 distinct ways to sum over the spin structures. Let the partition function of $\widetilde{\mathcal{T}}$ on a three-manifold with spin structure $s$ be $Z_s[\widetilde{\mathcal{T}}]$. The 16 distinct ways of summing over the spin structures correspond to weighing the sum by the partition function of $SO(r)_1$ with different $r$ mod 16:

$$Z[\mathcal{B}^{(r)}] = \sum_s Z_s[SO(r)_1] Z_s[\widetilde{\mathcal{T}}], \tag{4.10}$$

---

[6] We thank Nathan Seiberg for discussions on this point.

$$\begin{array}{ccc}
\widetilde{\mathcal{T}} & \xrightarrow{\;\;\sum_s\;\;} & \mathcal{B}^{(0)} \\
{\scriptstyle \times SO(r)_1}\Big\downarrow & & \Big\downarrow {\scriptstyle \frac{\times Spin(r)_1}{\mathbb{Z}_2}} \\
\widetilde{\mathcal{T}} \times SO(r)_1 & \xrightarrow{\;\;\sum_s\;\;} & \mathcal{B}^{(r)}
\end{array}$$

Figure 2: Starting from a spin TQFT $\widetilde{\mathcal{T}}$, we obtain 16 distinct non-spin TQFTs $\mathcal{B}^{(r)}$ from summing over the spin structures. The 16 $\mathcal{B}^{(r)}$ are related by (4.12).

where we denote the resulting non-spin TQFT as $\mathcal{B}^{(r)}$, with $\mathcal{B}^{(r+16)} = \mathcal{B}^{(r)}$. The non-spin TQFT $\mathcal{B}^{(r)}$ has an emergent anomalous $\mathbb{Z}_2$ one-form symmetry generated by a spin $\frac{1}{2}$ anyon [42].

How are the 16 non-spin TQFTs $\mathcal{B}^{(r)}$ related to each other? Let us first rewrite (4.10) as

$$Z[\mathcal{B}^{(r)}] = \sum_{b^{(2)}} \sum_{s_1, s_2} Z_{s_1}[SO(r)_1] Z_{s_2}[\widetilde{\mathcal{T}}] (-1)^{\int (s_1 - s_2) \cup b^{(2)}}, \tag{4.11}$$

where $b^{(2)}$ is a dynamical $\mathbb{Z}_2$ two-form gauge field coupled to the $\mathbb{Z}_2$ one-form connection $s_1 - s_2$.[7] The right hand side can be interpreted as summing over the spin structures in $SO(r)_1$ and $\widetilde{\mathcal{T}}$ separately, and coupling both of them to a dynamical $\mathbb{Z}_2$ two-form gauge field $b^{(2)}$. That is, we gauge the diagonal $\mathbb{Z}_2$ one-form symmetry. Hence, the non-spin TQFT $\mathcal{B}^{(r)}$ can be expressed as

$$\mathcal{B}^{(r)} \quad \longleftrightarrow \quad \frac{\mathcal{B}^{(0)} \times Spin(r)_1}{\mathbb{Z}_2^{(\mathbf{v})}}, \tag{4.12}$$

where the gauged $\mathbb{Z}_2^{(\mathbf{v})}$ one-form symmetry is generated by the tensor product of the spin $\frac{1}{2}$ lines of $\mathcal{B}^{(0)}$ and the spin $\frac{1}{2}$ line in the vector representation of $Spin(r)_1$. Importantly, this is generally a *different* gauging compared to the closed form expression (2.20) for the symmetry fractionalization $\mathbf{F}$: the latter involves the line in the spinor representation of $Spin(-4p)_1$. See Appendix C for more discussions on (4.12) versus the fractionalization map (2.20). We summarize the relation between the spin TQFT $\widetilde{\mathcal{T}}$ and the non-spin TQFTs $\mathcal{B}^{(r)}$ obtained from summing over the spin structures in Figure 2.

There is something special when $r = 8$. In $Spin(8)_1$, the $\mathbb{Z}_2$ lines in the vector and the two complex conjugate spinor representations all have spin $\frac{1}{2}$, and there is a $\mathbb{S}_3$ zero-form symmetry permuting them. Hence (4.12) for $r = 8$ coincides with (2.20) for $p = 2$, and we find that $\mathcal{B}^{(8)}$ is related to $\mathcal{B}^{(0)}$ by a fractionalization map $\mathbf{F}$. More generally, we have

$$\mathcal{B}^{(r+8)} \quad \longleftrightarrow \quad \mathbf{F}[\mathcal{B}^{(r)}], \tag{4.13}$$

where $\mathbf{F}$ uses the $\mathbb{Z}_2$ one-form symmetry generated by a line of spin $\frac{1}{2}$ (in the description (4.12) this line is the fermion line of $Spin(r)_1$ in the vector representation). For example, if $\widetilde{\mathcal{T}} = SO(0)_1$, then $\mathcal{B}^{(r)} = Spin(r)_1$. Indeed, $Spin(r)_1$ obeys (4.13) as discussed in (3.8).

Therefore we have shown that the 16 distinct non-spin TQFTs $\mathcal{B}^{(r)}$ are pairwise related by a fractionalization map $\mathbf{F}$. By Theorem 1, this implies that $\mathcal{B}^{(r+8)}$ and $\mathcal{B}^{(r)}$ are dual as spin TQFTs (4.1):

$$\mathcal{B}^{(r+8)} \times SO(8)_{-1} \quad \longleftrightarrow \quad \mathcal{B}^{(r)} \times SO(0)_1 \tag{4.14}$$

as spin TQFTs. The above also follows from (4.12) and the spin duality

$$Spin(r+8)_1 \times SO(8)_{-1} \quad \longleftrightarrow \quad Spin(r)_1 \times SO(0)_1, \tag{4.15}$$

---

[7]Note that the difference between any two spin structures is a $\mathbb{Z}_2$ gauge field.

where we have used the fact that $Spin(r)_1$ as a spin TQFT is dual to the fermionic $\mathbb{Z}_2$ gauge theory with level $-r \sim -(r+8)$ [31]. We conclude that, when viewed as spin theories, there are only 8 (instead of 16) distinct TQFTs from summing over the spin structures of a seed spin TQFT $\widetilde{\mathcal{T}}$.

These 8 spin TQFTs are obtained by gauging the zero-form symmetry $(-1)^F$ of $\widetilde{\mathcal{T}}$. Indeed, there are 8 distinct ways to gauge a $\mathbb{Z}_2$ zero-form symmetry in $(2+1)d$, classified by $\Omega^3_{spin}(B\mathbb{Z}_2) = \mathbb{Z}_8$ [43–46]. The difference between gauging $(-1)^F$ versus summing over the spin structures is that the former does not project out the transparent fermion line, while the latter does. Consequently, gauging $(-1)^F$ of a spin TQFT gives 8 distinct spin TQFTs, while summing over the spin structures of a spin TQFT gives 16 distinct non-spin TQFTs.

As discussed in [42], summing over the spin structures in $(2+1)d$ gives a non-spin theory with an anomalous $\mathbb{Z}_2$ 1-form symmetry generated by a fermion line. On the other hand, gauging $\mathbb{Z}_2$ $(-1)^F$ 0-form symmetry in $(2+1)d$ gives a dual $\mathbb{Z}_2$ 1-form symmetry that is non-anomalous i.e. the symmetry line is a boson. And one can gauge this 1-form symmetry to recover the original theory. This boson is the composite of the transparent fermion line present in spin TQFTs and the fermion line arises from summing over the spin structures.

Let us perform the gauging of $(-1)^F$ more explicitly. The $\mathbb{Z}_2$ spin SPT in $(2+1)d$ is classified by $r$ mod 8, whose partition function is [31]:

$$e^{-if_r[A^{(1)}]} = \frac{Z_{s+A^{(1)}}[SO(r)_1]}{Z_s[SO(r)_1]}, \tag{4.16}$$

where $A^{(1)}$ is the one-form background $\mathbb{Z}_2$ gauge field. Starting with a spin TQFT $\widetilde{\mathcal{T}}$, different ways of gauging $(-1)^F$ give 8 distinct spin TQFTs $\mathcal{F}^{(r)}$ whose partition functions are

$$Z[\mathcal{F}^{(r)}] = \sum_{a^{(1)}} Z_{s+a^{(1)}}[\widetilde{\mathcal{T}}] \frac{Z_{s+a^{(1)}}[SO(r)_1]}{Z_s[SO(r)_1]} = Z[\mathcal{B}^{(r)}] Z_s[SO(r)_{-1}], \tag{4.17}$$

where we sum over dynamical $\mathbb{Z}_2$ one-form gauge field $a^{(1)}$. That is, the 8 spin TQFTs $\mathcal{F}^{(r)}$ (obtained from gauging $(-1)^F$) are related to the 16 non-spin TQFTs $\mathcal{B}^{(r)}$ (obtained from summing over the spin structures) as

$$\mathcal{F}^{(r)} = \mathcal{B}^{(r)} \times SO(r)_{-1}. \tag{4.18}$$

Indeed, from (4.14) we have $\mathcal{F}^{(r+8)} = \mathcal{F}^{(r)}$.

Let us contrast these two operations in a specific example where $\widetilde{\mathcal{T}} = U(1)_1 = SO(2)_1$. The $(-1)^F$ symmetry is identified with the $\mathbb{Z}_2$ subgroup magnetic $U(1)$ zero-form symmetry [31, 41]. Summing over the spin structures (without tensoring additional $SO(r)_1$) gives the non-spin TQFT $\mathcal{B}^{(0)} = Spin(2)_1 = U(1)_4$. On the other hand, gauging the $(-1)^F$ zero-form symmetry (with trivial $3d$ fermionic $\mathbb{Z}_2$ SPT) corresponds to the following Lagrangian

$$\frac{1}{4\pi} ada + \frac{1}{2\pi} adb + \frac{2}{2\pi} bdc, \tag{4.19}$$

where $a, b, c$ are all dynamical $U(1)$ gauge fields. Here $a$ is the $U(1)$ gauge field for $\widetilde{\mathcal{T}} = U(1)_1$, $b$ is the dynamical gauge field for the $(-1)^F$ symmetry, and $c$ is a multiplier enforcing $b$ to be a $\mathbb{Z}_2$ gauge field. Let us do the following sequence of change of variables: first $a \rightarrow a - b$, then $b \rightarrow b + 2c$. The gauged theory is then recognized as $\mathcal{F}^{(0)} = U(1)_1 \times U(1)_{-1} \times U(1)_4 = SO(0)_1 \times Spin(2)_1$. The latter $\mathcal{F}^{(0)}$ is a spin TQFT which is different from the non-spin TQFT $\mathcal{B}^{(0)} = Spin(2)_1$ obtained from summing over the spin structures. See [20] for further examples of gauging the $(-1)^F$ symmetry of spin TQFTs (not to be confused with summing over the spin structures).

## 4.3 Implications for time-reversal symmetry

Let us discuss the implication of Lemma 1 and Theorem 1 for theories with time-reversal symmetry.

**Corollary 1** Let $\mathcal{T}$ be a non-spin QFT with framing anomaly $c$. Suppose it is time-reversal invariant as a spin QFT, then $c \in \mathbb{Z}$. Furthermore,

- If $c \notin 4\mathbb{Z}$, then the theory must have a $\mathbb{Z}_2$ Abelian anyon of spin $\frac{c}{4}$.

- $c \in 4\mathbb{Z}$ if and only if $\mathcal{T}$ is a time-reversal invariant non-spin QFT.

We prove the corollary in the following. Let $\mathcal{T}'$ (whose framing anomaly is $-c$ mod 8) be the time-reversal image of the non-spin QFT $\mathcal{T}$. By assumption, $\mathcal{T}$ and $\mathcal{T}'$ are dual as spin QFTs (4.1). By Lemma 1, $\Delta c = c - (-c) \in 2\mathbb{Z}$, *i.e.* $c \in \mathbb{Z}$.

The second and the third statements follow directly from Lemma 1 and Theorem 1. When $\Delta c = 2c \notin 8\mathbb{Z}$ the theory has a $\mathbb{Z}_2$ one-form symmetry generated by a line of spin $\frac{\Delta c}{8} = \frac{c}{4}$ by Lemma 1. By Theorem 1, $\mathcal{T} \leftrightarrow \mathcal{T}'$ if and only if $\Delta c = 2c \in 8\mathbb{Z}$.

As an application, consider a time-reversal (or particle-hole) invariant fermionic system in $(2+1)d$ (for instance, it can be a system of electrons). Suppose the system flows to the tensor product of a time-reversal symmetric fermionic theory and an extra emergent intrinsically bosonic system with $c \notin 4\mathbb{Z}$. Then the bosonic system must have an anomalous $\mathbb{Z}_2$ one-form symmetry generated by an Abelian anyon of spin $\frac{c}{4}$, or the time-reversal (particle-hole) symmetry must be spontaneously broken.

Examples of this are $QCD_3$ with fermions in the tensor representation as discussed in [31, 47–49]. Let us take for instance the UV theory to be $SU(N)_0$ with one massless adjoint Majorana fermion. We take $N$ to be even to avoid the standard parity anomaly, and the theory enjoys the time-reversal symmetry. The UV theory has a $\mathbb{Z}_2$ subgroup of the $\mathbb{Z}_N$ center one-form symmetry, whose 't Hooft anomaly can be computed by giving the fermion a large mass (while preserving the one-form symmetry). The resulting theory is $SU(N)_{\pm N/2}$ viewed as a spin TQFT and the $\mathbb{Z}_2$ symmetry line has spin $\pm\frac{N-1}{4}(N/2)^2$ (they differ by the transparent fermion line which is present in any spin theory). Thus for $N = 2$ mod 4 the UV theory $SU(N)_0$ with one massless adjoint fermion has an anomalous $\mathbb{Z}_2$ one-form symmetry generated by a semion (or anti-semion). The theory has $\mathcal{N} = 1$ supersymmetry that is expected to be spontaneously broken [50], and thus the IR theory is expected to contain a Goldstino, which is time-reversal invariant.

The theorem we proved implies that if the IR theory contains an extra non-spin QFT whose one-form symmetry matches that of the UV theory[8], and if it preserves the time-reversal symmetry, then the bosonic QFT must have framing anomaly

$$\frac{c}{4} = \frac{N-1}{4} \mod \frac{1}{2} \quad \text{for } N = 2 \text{ mod } 4 . \tag{4.20}$$

Namely, the framing anomaly must be an odd integer. In particular, such non-spin theory cannot be a non-chiral TQFT with vanishing framing anomaly. This is consistent with the proposal in [47] based on fermion/fermion dualities, where the TQFT $U(N/2)_{N/2,N}$ has framing anomaly $c = (N^2 + 4)/8$. It is an intrinsically non-spin TQFT when $N = 2$ mod 4 where $c$ is an odd integer, while for $N = 0$ mod 4 it is an intrinsically spin TQFT where $c$ is a half integer.

---

[8] Here we assume the absence of accidental $\mathbb{Z}_2$ one-form symmetries in the IR.

## 4.4 Level/rank dualities for non-spin TQFTs

The level/rank duality of Chern-Simons theories is usually phrased as the equivalence of two spin TQFTs [31, 51–53]. When the two TQFTs can also be formulated as non-spin theories, our Theorem 1 implies that they are related by a fractionalization map. Furthermore, when $\Delta c \in 8\mathbb{Z}$, the spin level/rank duality implies that the two TQFTs are also dual as non-spin theories.

For example, the spin level/rank dualities imply that the following non-spin Chern-Simons theories $\mathcal{T}_1, \mathcal{T}_2$ satisfy $\mathcal{T}_1 \longleftrightarrow \mathbf{F}[\mathcal{T}_2]$ where the map uses the center $\mathbb{Z}_2$ (subgroup) one-form symmetry:

$$\mathcal{T}_1 \longleftrightarrow \mathbf{F}[\mathcal{T}_2]:\quad
\begin{array}{|c|c|c|c|}
\hline
\mathcal{T}_1 & \mathcal{T}_2 & \text{condition} & \Delta c \\
\hline
U(N)_{K,K+N} & U(K)_{-N,-N-K} & \text{odd } N,K & NK+1 \\
SU(N)_K & U(K)_{-N} & \text{even } N & NK \\
SO(N)_K & SO(K)_{-N} & \text{even } N,K & NK/2 \\
Sp(N)_K & Sp(K)_{-N} & \text{any } N,K & 2NK \\
\hline
\end{array}\;. \qquad (4.21)$$

In special cases when $\Delta c \in 8\mathbb{Z}$, the Chern-Simons theories are also dual as non-spin theories:

$$\mathcal{T}_1 \longleftrightarrow \mathcal{T}_2:\quad
\begin{array}{|c|c|c|}
\hline
\mathcal{T}_1 & \mathcal{T}_2 & \text{condition} \\
\hline
U(N)_{K,K+N} & U(K)_{-N,-N-K} & NK = 7 \bmod 8 \\
SU(N)_K & U(K)_{-N} & \text{even } N; NK = 0 \bmod 8 \\
SO(N)_K & SO(K)_{-N} & \text{even } N,K; NK = 0 \bmod 16 \\
Sp(N)_K & Sp(K)_{-N} & NK = 0 \bmod 4 \\
\hline
\end{array}\;. \qquad (4.22)$$

The above list agrees and generalizes the results of [52]. Substituting the case $K = N$ in the above list we find the time-reversal invariant bosonic TQFTs as discussed in [52].[9]

# 5 Application II: Chern-Simons matter duality

The Lorentz symmetry fractionalization has a natural application to Chern-Simons matter dualities in $(2+1)d$. In many examples of dualities, the Lagrangian fields obey certain spin/charge relation. For example, bosons/fermions are even/odd under the $\mathbb{Z}_2$ center of the gauge group $G$, respectively. Therefore, despite the appearance of fermions in the Lagrangian, the gauge-invariant local operators are all bosonic and the theory can be formulated on manifolds without a choice of the spin structure. In these cases, the Chern-Simons matter dualities may be viewed as non-spin dualities.

More precisely, the gauge group and the Lorentz group has the following global structure

$$\frac{G \times Spin(3)_{\text{Lorentz}}}{\mathbb{Z}_2}, \qquad (5.1)$$

where $Spin(3)_{\text{Lorentz}}$ is the double-cover of the Lorentz group, and the $\mathbb{Z}_2$ is the diagonal subgroup of the center for $G$ and that for $Spin(3)_{\text{Lorentz}}$. In the ultraviolet QFT, there are generally fermion matter fields that transform non-trivially under the $\mathbb{Z}_2$ center of the gauge group $G$, so the latter is generally not a one-form symmetry. In the infrared phase when we give the fermion fields large masses, they decouple and the $\mathbb{Z}_2$ center of the gauge group does not act on any matter fields at low energies. Thus the low energy theory enjoys an emergent

---

[9]We remark that such a time-reversal symmetry often combines with a $\mathbb{Z}_2$ one-form symmetry into a 2-group (which is referred to as an $H^3$ obstruction in [54]).

$\mathbb{Z}_2$ one-forms symmetry[10] that is inherited from the center of the ultraviolet gauge group. The twisting (5.1) in the ultraviolet activates a non-trivial background for the two-form background field $B$ of the one-form symmetry, $B = w_2(SO(3)_{\text{Lorentz}})$, implementing the Lorentz symmetry fractionalization.

Let us consider the following boson/fermion duality [55, 56]:

$$U(1)_2 + \phi \quad \longleftrightarrow \quad U(1)_{-\frac{3}{2}} + \psi. \tag{5.2}$$

The left hand side is manifestly a bosonic theory which does not require a choice of the spin structure. On the right hand side, the fermion $\psi$ has charge $+1$ and obeys the spin/charge relation with respect to the dynamical $U(1)$ gauge field, so the right theory is also bosonic. Therefore, (5.2) can be viewed as a non-spin duality.

By turning on a positive mass square for the boson, the left hand side is gapped to the $U(1)_2$ Chern-Simons theory. This relevant deformation corresponds to turning on a negative mass for the fermion, which would naively drive the right hand side to the $U(1)_{-2}$ Chern-Simons theory. However, as we discussed above, the $U(1)$ gauge group bundle on the right is twisted in the ultraviolet, which results in a Lorentz symmetry fractionalization (2.1). Consequently, the right hand side at long distance is actually $\mathbf{F}[U(1)_{-2}] = U(1)_2$, which correctly matches the left hand side when viewed as non-spin TQFTs.[11]

Another class of examples is the following infinitely many boson/fermion dualities:

$$Sp(N)_k + N_f \ \phi \ \text{in} \ \mathbf{2N} \quad \longleftrightarrow \quad Sp(k)_{-N+\frac{N_f}{2}} + N_f \ \psi \ \text{in} \ \mathbf{2k}. \tag{5.3}$$

The dualities (5.3) were proposed in [52] as between spin theories. Since the theories with fermion fields satisfy the spin/charge relation with respect to the dynamical gauge field, (5.3) can further be viewed as dualities for non-spin theories. Again we turn on a positive mass square for the boson on the left, which corresponds to the a negative mass for the fermion on the right. At long distance the non-spin TQFT on the left is the $Sp(N)_k$ Chern-Simons theory, while that on the right it is $\mathbf{F}[Sp(k)_{-N}]$ due to the twisted gauge bundle in the ultraviolet. Indeed, $Sp(N)_k \longleftrightarrow \mathbf{F}[Sp(k)_{-N}]$ as discussed in (4.22).

Similar to [52], the duality (5.3) describes a single bosonic phase transition for $N_f \leq N$, while for larger values of $N_f$ there are multiple phase transitions such as in [57] (see also [58] for other scenarios for large $N$). The phase transitions here are purely bosonic, in contrast to those in [52] where additional fermionic lines were involved.

# Acknowledgement

We thank Thomas Dumitrescu, Anton Kapustin, Zohar Komargodski, Nathan Seiberg, and Ryan Thorngren for discussions. We thank Maissam Barkeshli, Nathan Seiberg, and Zhenghan Wang for comments on a draft. S.H.S. would like to thank Nathan Seiberg for enlightening conversations on spin and non-spin TQFTs that inspired part of this work. The work of P.-S. H. is supported by the U.S. Department of Energy, Office of Science, Office of High Energy Physics, under Award Number DE-SC0011632, and by the Simons Foundation through the Simons Investigator Award. The work of S.H.S. is supported by the National Science Foundation grant PHY-1606531, the Roger Dashen Membership, and a grant from the Simons Foundation/SFARI (651444, NS). This work was performed in part at Aspen Center for Physics, which is supported by National Science Foundation grant PHY-1607611.

---

[10]The one-form symmetry can also be ruined by monopole operators, but this does not arise in the examples we discuss below.

[11]If instead we turn on a negative mass square for the boson and a positive mass for the fermion, then both sides are trivially gapped at long distance.

## A  $\mathbb{Z}_N$ gauge theories

The TQFT $(\mathcal{Z}_N)_K$ can be realized as the following $U(1) \times U(1)$ Chern-Simons theory [59–61]:

$$(\mathcal{Z}_N)_K : \quad \int \left( \frac{K}{4\pi} ada + \frac{N}{2\pi} adb \right). \tag{A.1}$$

For even $K$ the theory is non-spin and is the Dijkgraaf-Witten theory [24]. We have the identification $(\mathcal{Z}_N)_K = (\mathcal{Z}_N)_{K+2N}$. The line operators are $W_{n_e,n_m} = \exp\left[ in_e \oint a + in_m \oint b \right]$, labeled by an electric charge $n_e \in \mathbb{Z}$ and a magnetic charge $n_m \in \mathbb{Z}$. We will call $W_{n_e,0}$ the electric lines and $W_{0,n_m}$ the magnetic lines. The spin of $W_{n_e,n_m}$ is

$$h_{n_e,n_m} = \frac{n_e n_m}{N} - \frac{K n_m^2}{2N^2}. \tag{A.2}$$

The lines $W_{n_e,n_m}$ and $W_{n_e+N,n_m}$ are identified. For even $K$, the lines $W_{n_e,n_m}$ and $W_{n_e+K,n_m+N}$ are further identified, and we are left with $N^2$ lines.

## B  Duality map and the one-form symmetry

Consider two non-spin TQFTs $\mathcal{T}_1$ and $\mathcal{T}_2$ that are dual as spin TQFTs as in (4.1). In Lemma 1 of Section 4, we showed that if $\Delta c \notin 8\mathbb{Z}$, then $\mathcal{T}_1, \mathcal{T}_2$ must have a $\mathbb{Z}_2$ one-form symmetry. In this appendix, we show that the same is true even when $\Delta c \in 8\mathbb{Z}$ provided there is a nontrivial duality map between the spin theories.

Let $g^{\text{spin}}$ be the duality map that maps the anyons of $\mathcal{T}_1 \times SO(0)_1$ to those of $\mathcal{T}_2 \times SO(L)_1$. The duality map $g$ preserves the fusion, braiding, and all other correlation functions.

Given two non-spin TQFTs $\mathcal{T}_1, \mathcal{T}_2$ that are dual as spin theories, the duality map $g^{\text{spin}}$ in (4.1) might not be unique. Below we show that if there exits a duality map in (4.1) that mixes the lines in the non-spin TQFTs with the transparent fermion line (*i.e.* if $g^{\text{spin}}$ does not just map anyons of $\mathcal{T}_1$ to those of $\mathcal{T}_2$), then the theory $\mathcal{T}_1, \mathcal{T}_2$ must have a $\mathbb{Z}_2$ one-form symmetry. This is true even when $\Delta c \in 8\mathbb{Z}$: in such cases the $\mathbb{Z}_2$ one-form symmetry is generated by a line of integer spin.

In the spin duality (4.1), let us assume that there is a line $x \in \mathcal{T}_1$ that is mapped to a tensor product of a line $y \in \mathcal{T}_2$ and the transparent line $f$ under the duality map $g^{\text{spin}}$:

$$g^{\text{spin}} : \quad x \quad \longrightarrow \quad y \otimes f. \tag{B.1}$$

After summing over the spin structures, the duality map $g^{\text{spin}}$ turns into a duality map $g^{\text{non−spin}}$ between the two non-spin TQFTs in (4.3). The action of $g^{\text{non−spin}}$ on the anyons in $\mathcal{T}_1$ follows from (B.1), where the transparent fermion line $f$ is now in $(\mathcal{Z}_2)_0$ and $Spin(L)_1$ on the two sides.

How does $g^{\text{non−spin}}$ act on the new lines of spin 0 from $(\mathcal{Z}_2)_0$ that are introduced from summing over the spin structures? For example, consider the image of the electric line $e \in (\mathcal{Z}_2)_0$ on the left hand side of (4.3). The image must involve lines that are not present in $\mathcal{T}_2 \times SO(L)_1$, so it must be of the following form:

$$g^{\text{non−spin}} : \quad e \quad \longrightarrow \quad s \otimes e', \tag{B.2}$$

where $e'$ is a new line of spin $\frac{L}{16} \in \mathbb{Z}$ in $Spin(L)_1$ that generates a $\mathbb{Z}_2$ one-form symmetry. Here $s \in \mathcal{T}_2$ has integer spin and is a symmetry line (that can be trivial). Next, we braid the lines in (B.1) and (B.2). Since $e, x$ belong to different parts in the tensor product theory $\mathcal{T}_1 \times (\mathcal{Z}_2)_0$ they

have trivial braiding. On the other hand, $e', f \in Spin(L)_1$ braids non-trivially. This implies $s$ must braid non-trivially with $y$, and in particular $s$ cannot be the trivial line in $\mathcal{T}_2$. Thus we conclude $\mathcal{T}_2$ must have a $\mathbb{Z}_2$ one-form symmetry generated by the line $s$ of integer spin.

We remark that if a theory $\mathcal{T}_2$ has a $\mathbb{Z}_2$ one-form symmetry generated by a line of integer spin, then as a spin TQFT it has a $\mathbb{Z}_2$ ordinary symmetry that mixes the lines in $\mathcal{T}_2$ with the transparent fermion line $f$: for lines $y \in \mathcal{T}_2$ odd under this $\mathbb{Z}_2$ one-form symmetry, the ordinary symmetry changes its type into the product of the fusion $y \cdot s \cdot f$.

## C  Summing over the spin structures with invertible spin TQFTs in (1+1)d and (2+1)d

Consider two spin theories $\widetilde{\mathcal{T}}_1$ and $\widetilde{\mathcal{T}}_2$ differ by an invertible spin TQFT, which can be described by the $SO(r)_1$ Chern-Simons theory. Let $\mathcal{T}_1, \mathcal{T}_2$ be the non-spin theories obtained by summing over the spin structures of $\widetilde{\mathcal{T}}_1$ and $\widetilde{\mathcal{T}}_2$, respectively.[12] Both $\mathcal{T}_1$ and $\mathcal{T}_2$ has an emergent anomalous $\mathbb{Z}_2$ one-form symmetry generated by a spin $\frac{1}{2}$ fermion line [42]. The discussion in Section 4.2 implies that:

$$\mathcal{T}_1 \quad \longleftrightarrow \quad \frac{\mathcal{T}_2 \times Spin(r)_1}{\mathbb{Z}_2^{(\mathbf{v})}} \,, \tag{C.1}$$

where the gauged $\mathbb{Z}_2^{(\mathbf{v})}$ one-form symmetry is generated by the tensor product of the fermion line in $\mathcal{T}_2$ and the fermion line of $Spin(r)_1$ in the vector representation.[13]

The right hand side of (C.1) can also be interpreted as gauging a zero-form $SO(r)$ symmetry of $\mathcal{T}_2$, as we explain in the following. We first activate a two-form background gauge field for the $\mathbb{Z}_2$ one-form symmetry (generated by the fermion) using $w_2(SO(r))$ of the background $SO(r) = Spin(r)/\mathbb{Z}_2^{(\mathbf{v})}$ bundle. Next, we add a Chern-Simons term $SO(r)_1$ as a local counterterm for the $SO(r)$ background gauge field. Finally, we promote the $SO(r)$ gauge fields to be dynamical. The resulting non-spin theory is the right hand side of (C.1):

$$\mathcal{T}_1 \quad \longleftrightarrow \quad \mathcal{T}_2 \text{ w/ gauging } SO(r) \text{ symmetry } . \tag{C.2}$$

Note that the transformation (C.1) is different from the fractionalization map in (2+1)$d$ (2.20):

$$\mathbf{F}: \quad \mathcal{T} \quad \longrightarrow \quad \frac{\mathcal{T} \times Spin(4p)_{-1}}{\mathbb{Z}_2^{(\mathbf{s})}} \,, \tag{C.3}$$

where the gauged $\mathbb{Z}_2^{(\mathbf{s})}$ one-form symmetry on the right hand side is generated by the tensor product of the spin $\frac{p}{4}$ line of $\mathcal{T}$ and the spin $-\frac{p}{4}$ line in the spinor representation of $Spin(4p)_{-1}$. The two-form background gauge field for the $\mathbb{Z}_2$ one-form symmetry of $\mathcal{T}$ is activated by the $w_2(Spin(4p)/\mathbb{Z}_2^{(\mathbf{s})})$ of the $Spin(4p)/\mathbb{Z}_2^{(\mathbf{s})} = Ss(4p)$ gauge bundle. Thus the fractionalization map in $(2+1)d$ can be interpreted as gauging an $Ss(4p)$ symmetry (with local counterterm given by the level-one Chern-Simons term):

$$\mathbf{F} \text{ in } (2+1)d: \quad \mathcal{T} \quad \longrightarrow \quad \mathcal{T} \text{ w/ gauging } Ss(4p) \text{ symmetry } . \tag{C.4}$$

The relation (C.1) has a counterpart in $(1+1)d$, where the invertible spin TQFT is the Arf invariant of the spin two-manifold. The action of this invertible spin TQFT is $(-1)^{\mathrm{Arf}[s]}$, where

---

[12]Theories with a tilde sign are spin, while those without are non-spin.
[13]We thank Ryan Thorngren for discussions.

$$\widetilde{\mathcal{T}}_1^{2d} \xrightarrow{\ \sum_s\ } \mathcal{T}_1^{2d} \qquad\qquad \widetilde{\mathcal{T}}_1^{3d} \xrightarrow{\ \sum_s\ } \mathcal{T}_1^{3d}$$

left diagram: vertical arrows labeled $\times(-1)^{\text{Arf}}$ and gauge $\mathbb{Z}_2$; right diagram: vertical arrows labeled $\times SO(r)_1$ and gauge $SO(r)$

$$\widetilde{\mathcal{T}}_2^{2d} \xrightarrow{\ \sum_s\ } \mathcal{T}_2^{2d} \qquad\qquad \widetilde{\mathcal{T}}_2^{3d} \xrightarrow{\ \sum_s\ } \mathcal{T}_2^{3d}$$

Figure 3: The relation between the two spin theories $\widetilde{\mathcal{T}}_{1,2}$ and the two non-spin theories $\mathcal{T}_{1,2}$ in $(1+1)d$ (left) and in $(2+1)d$ (right). Compared to the main text, the superscripts "3d" are added for clarity. In the figure on the left, $(-1)^{\text{Arf}}$ can be represented by a $(1+1)d$ spin $\mathbb{Z}_2$ gauge theory with action given by the right hand side of (C.5).

Arf$[s] = 0$ if $s$ is an even spin structure and $1$ if $s$ is odd. Analogous to the $SO(r)_1$ Chern-Simons theory in $(2+1)d$, the $(1+1)d$ invertible spin TQFT can be alternatively described by a $\mathbb{Z}_2$ gauge theory coupled to the Arf invariant as follows:

$$(-1)^{\text{Arf}[s]} = \frac{1}{2^g} \sum_{a^{(1)}} (-1)^{\text{Arf}[s+a^{(1)}]+\text{Arf}[s]}, \tag{C.5}$$

where $a^{(1)}$ is the dynamical $\mathbb{Z}_2$ one-form gauge field and $g$ is the genus of the two-manifold. In other words, this fermionic $\mathbb{Z}_2$ gauge theory can be obtained by gauging a $\mathbb{Z}_2$ symmetry of the $(1+1)d$ fermionic SPT phase $(-1)^{\text{Arf}[s+a^{(1)}]+\text{Arf}[s]}$ of $\Omega_{spin}^2(B\mathbb{Z}_2) = \mathbb{Z}_2 \times \mathbb{Z}_2$ (which does not come from $\Omega_{spin}^2(pt) = \mathbb{Z}_2$) [46]. Now consider two spin theories $\widetilde{\mathcal{T}}_1^{2d}$ and $\widetilde{\mathcal{T}}_2^{2d}$ differ by this invertible spin TQFT. Sometimes we can sum over the spin structures of $\widetilde{\mathcal{T}}_1^{2d}$ and $\widetilde{\mathcal{T}}_2^{2d}$, to obtain two non-spin theories $\mathcal{T}_1^{2d}$ and $\mathcal{T}_2^{2d}$. When this is the case, there is an emergent non-anomalous $\mathbb{Z}_2$ zero-form symmetry in both $\mathcal{T}_1$ and $\mathcal{T}_2$. The two non-spin theories $\mathcal{T}_1^{2d}$ and $\mathcal{T}_2^{2d}$ are related by a $\mathbb{Z}_2$ orbifold [42, 62–67]:

$$(1+1)d: \quad \mathcal{T}_1^{2d} \quad \longleftrightarrow \quad \mathcal{T}_2^{2d} \text{ w/ gauging } \mathbb{Z}_2 \text{ symmetry}. \tag{C.6}$$

We compare (C.6) in $(1+1)d$ with (C.2) in $(2+1)d$ in Figure 3.

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
