# Peer review of "Lorentz Symmetry Fractionalization and Dualities in (2+1)d"

_SciPost Physics, doi:SciPost Phys. 8, 018 (2020)_

## Round 1 · Referee Report · Anonymous (Referee 1) · 2020-1-6

Report

This paper introduces a fractionalization map, and relates this map to the symmetry fractionalization of the Lorentz group, the 1-form Z_2 symmetry anomaly. The paper also proves Lemma 1 and Theorem 1, and briefly comment on the implications for time-reversal symmetry.

Referee will be happy to recommend the paper for publication as long as the authors can answer/resolve Referee's questions/comments:

1) The authors focus on only the 1-form Z_2 symmetry. Referee wonders whether there can be any generalization of 1-form Z_n or 1-form U(1) electric or 0-form magnetic symmetry story for the fractionalization map? If so, can the authors make some comments on this generalization?

2) Also, what is the validity of the main Lemma 1, Theorem 1 and Collorary 1? It looks that in Abstract it says a stronger claim for non-spin QFT and spin QFT; while in the main text, it says the more restricted non-spin TQFT and spin TQFT. Some clarifications are required.

3) The authors define a fractionalization map, which can map a QFT to another QFT. However, the original QFT is known to be well-defined (for example, the bosonic TQFT needs to be given by data of modular tensor category). But how do we know whether the output data is also a well-defined QFT?

4) Is the essence of framing anomaly, simply equal to the chiral central charge c mod 8?

The entries given in eq.(2.6) and eq.(2.7) on the q charges are related to the modular S matrix of SL(2,Z). Maybe the authors can add a comment on this.

5) Suppose the time-reversal symmetries are involved, such that we require the time-reversal quantum number for eq.(2.6) and eq.(2.7), etc. Do we have a generalized fractionalization map? Would this be a case for lifting Spin to Pin^+ and Pin^-, or are there something more than that for a generalized fractionalization map? (p.s. there are some generalization of Lemma 1 and Theorem 1 in Sec 4.3 Implications for time-reversal symmetry, but it may be useful to know easier examples.)

6) In p.8, "In the case of the Lorentz symmetry fractionalization (2.1), the one-form symmetry anomaly gives rise to the framing anomaly. Here we discuss the change of the framing anomaly under the fractionalization map."

1-form symmetry can induce framing anomaly, by requiring, for example, B= \pi w_2. and then P(B) \sim P(w_2) \sim RR which produces the framing anomaly.

But does a nonzero 1-form symmetry anomaly imply a framing anomaly (chiral central charge c mod 8), in 3d? Is this always true?

7) In eq (2.21), the authors claimed that when p=0 mod 4, F[T]<->T. As said in the introduction (on the top of page 3), the author claimed that F, in this case, is a 0-form symmetry of the theory. However, this is clearly not an automorphism of the lines (which is usually discussed in the anyon theory, e.g. https://arxiv.org/abs/1410.4540) because the topological spin of the lines are not preserved by F. The authors may need to clarify more about the differences between the usual 0-form symmetry in the anyon theory (realized as an automorphism of the lines) and the F map.

8) In p.13, "In the generalization to bosonic quantum field theory, the framing anomaly in the following discussions can be defined by the coefficient of the parity-odd contact term in the stress tensor two-point function."

In this definition, can the framing anomaly still equal to the chiral central charge c mod 8? Is the parity-odd contact term only a mod 2 class or mod N for what N?

9) In eq.(4.4), there is a list of 1-form symmetry of Spin(M)1. Does this 1-form G symmetry also depend on the level-k of Spin(M)k? What is the dependence of k and the G?

10) In p.17, "The difference between gauging (−1)^F versus summing over the spin structures is that the former does not project out the transparent fermion line, while the latter does. Consequently, gauging (−1)^F of a spin TQFT gives 8 distinct spin TQFTs, while summing over the spin structures of a spin TQFT gives 16 distinct non-spin TQFTs. "

Referee does not fully grasp this meaning and gets confused. Shouldn't the gauging (−1)^F be equivalent to a bosonization procedure? Shouldn't sum over the spin structures also a bosonization procedure? Shouldn't both procedures term a spin QFT to a non-spin QFT?

If (−1)^F is gauged, then there is no fermion parity symmetry and thus no (−1)^F symmetry, shouldn't the theory be bosonic and non-spin? How is this procedure related to "fermionic SPT phases in higher dimensions and bosonization in https://arxiv.org/abs/1701.08264"?

11) Referee finds some recent or previous works relevant to this Lorentz symmetry fractionalization (with or without time-reversal on non-spin manifold) can be useful for readers:

https://arxiv.org/abs/1911.00589 https://arxiv.org/abs/1712.08639 https://arxiv.org/abs/1711.11587

12) In p.16 the the Z_2 lines --> typo.

13) In p.13, in Lemma 1, to identify the 1-form Z_2 symmetry, do the 1-form symmetry matched for both sides of eq (4.1) for all L of SO(L)_1? Does \pi_1(SO(L)) = Z_2 for L >2 correspond to any symmetry? Say, 1-form symmetry or 0-form magnetic symmetry in 3d?

14) Lastly and importantly, given a gauge theory of a gauge group G_{gauge}, is this correct that we can determine that 1-form electric symmetry by looking at the two data: center G_{gauge} and the matter field representation? Are there other hidden 1-form symmetry not given by the data, but depending on the Chern-Simons level? How about the 0-form magnetic symmetry?

  • validity: high
  • significance: high
  • originality: high
  • clarity: high
  • formatting: excellent
  • grammar: excellent

Author:  Po-Shen Hsin  on 2020-01-08  [id 700]

(in reply to Report 1 on 2020-01-06)
Category:
answer to question

1) For 1-form symmetry A in d spacetime dimension, the fractionalization maps correspond to the elements in H^2(SO(d),A). For A=Z_n there is nontrivial map for even n and only trivial map for odd n, and similarly A=U(1) has a nontrivial map.

For 0-form symmetry A^{(0)} a generalized fractionalization map can be defined for time-reversal invariant theories, given by H^1(O(d),A^{(0)}) i.e. the map changes how time-reversal symmetry acts on the operators odd under Z_2 subgroups of the 0-form symmetry A^{(0)}.

For q-form symmetry A^{(q)} the generalized fractionalization map activates the background for the q-form symmetry given by the pullback to the spacetime of the elements in H^{q+1}(SO(d), A^{(q)}), or more generally H^{q+1}(O(d), A^{(q)}) when there is time-reversal symmetry.

2) They are valid for general non-spin QFTs (explained in the second paragraph of section 4). We will rephrase the word TQFT to be QFT in the theorems.

3) The fractionalization map activates a background for the global symmetry in a well-defined QFT, thus it gives a well-defined QFT (the theory can have an 't Hooft anomaly as in general QFTs).

4) The framing anomaly c can be understood as the chiral central charge c mod 8. The braiding of symmetry line a and general line b can be expressed as S_{a,b}/S_{0,b} where 0 denotes the trivial line and S is the modular S matrix.

5) Yes. For instance, consider the time-reversal symmetry that does not permute the lines in the untwisted Z_2 gauge theory. There is a generalized fractionalization map given by activating the following background for the Z_2 1-form symmetry generated by the f line B = w1^2. This map changes the e and m particles (that are charged under the 1-form symmetry) to be both Kramers doublet. One can also consider Pin^\pm(d) bundles (that are special cases of O(d) bundles), where for Pin^+ (Pin^-) one sets w2=0 (w2+w1^2=0) so there are fewer distinct generalized fractionalization maps.

6) A theory with an anomalous 1-form symmetry does not necessarily have a framing anomaly. For instance, in the twisted Z_2 gauge theory the Z_2 1-form symmetry generated by the semion is anomalous (since it is not a boson), but the Z_2 gauge theory itself has trivial framing anomaly.

7) Unlike unitary symmetry, an antiunitary symmetry satisfies (72) of https://arxiv.org/pdf/1410.4540.pdf with complex conjugation. The introduction refers to the example in section 2.5 of twisted Z_2 gauge theory where the map F with respect to the boson is a time-reversal symmetry.

8) The contact term odd under parity transformation also corresponds to the gravitational Chern-Simons term (or tr RR term in 4d bulk) and it is related to the chiral central charge. In bosonic theory this contact term has ambiguity of mod 16 in the normalization of https://arxiv.org/pdf/1206.5218.pdf coming from adding properly-quantized bosonic gravitational Chern-Simons term. In the discussion of section 4 this coefficient is identified with 2c.

9) The 1-form symmetry group of Spin(M)_k is given by the center of Spin(M) and it does not depend on the level k.
The level k only affects the 't Hooft anomaly of the 1-form symmetry i.e. the spin of the symmetry lines. For odd M the generator for the Z_2 1-form symmetry has spin k/2, for M=0 mod 4 the extra Z_2 generator has spin MK/16, and for M=2 mod 4 the generator for Z_4 1-form symmetry has spin MK/16.

10) The difference between gauging (-1)^F 0-form symmetry and summing over spin structure is that the former retains the transparent fermion line that is present in any spin QFT, while the later does not. Thus the former gives a spin QFT, while the later gives a non-spin QFT. This difference also explains the following properties: As discussed in https://arxiv.org/pdf/1605.01640.pdf and https://arxiv.org/abs/1701.08264, summing over the spin structures in (2+1)d gives a non-spin theory with an anomalous Z2 1-form symmetry generated by a fermion line. On the other hand, gauging Z_2 (-1)^F 0-form symmetry in (2+1)d gives a dual Z_2 1-form symmetry that is non-anomalous i.e. the symmetry line is a boson. And one can gauge this 1-form symmetry to recover the original theory. This boson is the composite of the transparent fermion line present in spin TQFTs and the fermion line arises from summing over the spin structures.

11) Thanks.

12) Thanks.

13) SO(L)_1 for every L only has the identity line and the transparent fermion line (the Wilson line in the vector representation). Thus the theories for all L have the same 1-form symmetry. pi_1(SO(L))=Z_2 for L>2 is the Z_2 magnetic 0-form symmetry in (2+1)d and it can be identified with (-1)F symmetry in SO(L)1 theory.

14) When a continuous gauge group has non-trivial pi_1, the electric 1-form symmetry also depends on the Chern-Simons level. In such case there are monopole operators that end on suitable Wilson lines depending on the Chern-Simons level and reduces the electric 1-form symmetry to be a subgroup. An example of this is U(1)_k with Z_k 1-form symmetry instead of U(1). There can also be hidden 1-form symmetry not from the center of the gauge group and depends on the Chern-Simons level. An example is O(n) Chern-Simons theory discussed in https://scipost.org/10.21468/SciPostPhys.4.4.021. The 0-form magnetic symmetry is given by pi_1 of the gauge group if there is no magnetic matter.

---

## Editorial Decision

published